# Multi-Omics Approaches for Liver Reveal the Thromboprophylaxis Mechanism of Aspirin Eugenol Ester in Rat Thrombosis Model

**DOI:** 10.3390/ijms25042141

**Published:** 2024-02-10

**Authors:** Qi Tao, Ning Ma, Liping Fan, Wenbo Ge, Zhendong Zhang, Xiwang Liu, Jianyong Li, Yajun Yang

**Affiliations:** 1Key Lab of New Animal Drug Project of Gansu Province, Key Lab of Veterinary Pharmaceutical Development of Ministry of Agriculture and Rural Affairs, Lanzhou Institute of Husbandry and Pharmaceutical Sciences of CAAS, Lanzhou 730050, China; taoqi19951224@163.com (Q.T.); fanliping43t@163.com (L.F.); gewb1993@163.cm (W.G.); 13027721013@163.com (Z.Z.); xiwangliu@126.com (X.L.); 2College of Veterinary Medicine, Hebei Agricultural University, Baoding 071001, China; maning9618@163.com

**Keywords:** aspirin eugenol ester (AEE), thrombosis, transcriptomics, proteomics, metabolomics

## Abstract

Aspirin eugenol ester (AEE) is a novel medicinal compound synthesized by esterifying aspirin with eugenol using the pro-drug principle. Pharmacological and pharmacodynamic experiments showed that AEE had excellent thromboprophylaxis and inhibition of platelet aggregation. This study aimed to investigate the effect of AEE on the liver of thrombosed rats to reveal its mechanism of thromboprophylaxis. Therefore, a multi-omics approach was used to analyze the liver. Transcriptome results showed 132 differentially expressed genes (DEGs) in the AEE group compared to the model group. Proteome results showed that 159 differentially expressed proteins (DEPs) were identified in the AEE group compared to the model group. Six proteins including fibrinogen alpha chain (Fga), fibrinogen gamma chain (Fgg), fibrinogen beta chain (Fgb), orosomucoid 1 (Orm1), hemopexin (Hpx), and kininogen-2 (Kng2) were selected for parallel reaction monitoring (PRM) analysis. The results showed that the expression of all six proteins was upregulated in the model group compared with the control group. In turn, AEE reversed the upregulation trend of these proteins to some degree. Metabolome results showed that 17 metabolites were upregulated and 38 were downregulated in the model group compared to the control group. AEE could reverse the expression of these metabolites to some degree and make them back to normal levels. The metabolites were mainly involved in metabolic pathways, including linoleic acid metabolism, arachidonic acid metabolism, and the tricarboxylic acid (TCA) cycle. Comprehensive analyses showed that AEE could prevent thrombosis by inhibiting platelet activation, decreasing inflammation, and regulating amino acid and energy metabolism. In conclusion, AEE can have a positive effect on thrombosis-related diseases.

## 1. Introduction

Thrombosis can obstruct or block blood flow, as well as cause serious complications such as ischemic stroke [1] and myocardial infarction [2]. Hypercoagulability, blood disorder, and injury to the blood vessel wall are the main causes of thrombosis [3]. Thrombosis and its related disorders are the leading cause of morbidity and mortality worldwide [4]. Fortunately, a number of thromboprophylaxis agents, such as aspirin (acetylsalicylic acid, ASA), heparin, and clopidogrel, are available for the prevention of thrombosis in the current pharmaceutical market. However, recent studies showed that these agents, especially in long-term use, had many inescapable adverse effects, such as gastrointestinal bleeding, thrombocytopenia, and damage to liver and kidney function [5,6]. It is significant to develop alternative thromboprophylaxis agents with high efficacy and low toxicity.

As a cyclooxygenase inhibitor, aspirin has inhibitory platelet activation and anti-inflammatory effects and is commonly used in the prevention of thrombosis [7,8,9]. Eugenol, a natural product mainly derived from several plants, including cinnamon, cloves, and bay leaves, has anti-inflammatory, antipyretic, thromboprophylaxis, antioxidant, and analgesic properties [10,11,12]. Aspirin and eugenol have similar pharmacologic profiles. However, the gastrointestinal side effects of aspirin and the chemical structure instability of eugenol limit their applications. Based on the pro-drug principle, aspirin eugenol ester (AEE) was synthesized by esterification reaction [13]. Through the chemical masking of the hydroxyl group and the carboxyl group, AEE increases the structural stability and reduces the gastrointestinal side effects of its precursors [14]. AEE was metabolized into salicylic acid and eugenol in vivo [15], which exerted synergistic action to increase the therapeutic effects.

As high-throughput methods, omics techniques, including transcriptomics, proteomics, and metabolomics, are effective ways to investigate the action mechanism of compounds in biological systems [16,17]. Transcriptomics gives an overview of gene expression patterns, and proteomics looks into the production of proteins and their interactions. With the analysis of low molecular weight metabolites, metabolomics provided information to understand the metabolic responses of living systems to pathophysiological stimuli. Single-omics techniques only convey limited information at the single level of genes, proteins, or metabolites. Integrative analysis of transcriptomics, proteomics, and metabolomics data can provide global information on biological systems and is helpful in finding the interaction networks, modified pathways, and changing processes. Recently, the combination of omics methods has been well utilized in the basic research of thrombosis, such as pathogenesis, biomarker discovery, and new drug development.

The rat thrombosis model induced by carrageenan is widely used to assess the thromboprophylaxis effect of compounds, natural products, and enzymes [18,19,20]. With the application of this model, thromboprophylaxis properties of AEE were found to reduce fibrinogen levels, inhibit platelet aggregation, and regulate the hemorheological parameters and coagulation function [21,22,23]. The liver, as an important organ in the body, performs essential functions related to protein synthesis, metabolism, and immunity [24], yet the effect of AEE on the liver in thrombosis rats has not been systematically studied. Transcriptomics based on high-throughput sequencing, proteomics analysis by isobaric tags for relative and absolute quantitation (iTRAQ), and metabonomics analysis by untargeted profiling of liquid chromatography–mass spectrometry (LC-MS) spectra were conducted to investigate the different genes, proteins, and metabolites profiles in the liver of thrombosis rats. This study aims to better understand the antithrombosis mechanism action of AEE by multi-omics approaches in the liver and to provide theoretical support and ideas for AEE in the treatment of thrombosis-related diseases.

## 2. Results

### 2.1. Transcriptome Sequencing and Gene Expression

#### 2.1.1. Transcriptome Sequencing and Identification of DEGs

Appendix A lists the summary of the sequencing reads. After removing the low-quality reads, clean reads from the liver samples were obtained, and the proportion of clean reads was greater than 96%, indicating that the sequencing data had good reliability and quality. The transcriptome profiles of the samples were revealed by PCA that the samples in the control and the model were separately clustered, and the samples in AEE, ASA, and Eug groups exhibited a tendency to be away from those in the model (Figure 1A), indicating distinct expression profiles in livers among the different groups. DEGs were screened with fold change > 2 and *p*-value < 0.05. In Figure 1B, compared with the control group, 556 genes were differentially expressed in the model group, of which 254 genes were upregulated and 302 were downregulated. Meanwhile, 242, 94, and 132 DEGs were identified in the Eug, ASA, and AEE groups, respectively. As shown in the Venn diagram (Figure 1B), 16 DEGs were overlapped in all groups, and 438, 48, 29, and 132 DEGs were specifically identified in the control, AEE, ASA, and Eug groups, respectively.

#### 2.1.2. Functional Enrichment of DEGs

KEGG pathway enrichment analysis was used to reveal the biological roles of the DEGs. The top 20 key KEGG pathways are shown in Figure 1C–F. In the model group, the DEGs were mainly related to the JAK-STAT signaling pathway, complement and coagulation cascades, arachidonic acid metabolism, glycerophospholipid metabolism, the NF-κB signaling pathway, and the TNF signaling pathway. DEGs in the ASA group were found to be associated with the metabolism of linoleic acid, arachidonic acid, and amino acids. In the Eug group, the DEGs were involved in the mTOR signaling pathway, the Toll-like receptor signaling pathway, and the IL-17 signaling pathway. In the AEE group, the DEGs were significantly enriched in pathways such as the IL-17 signaling pathway, the MAPK signaling pathway, the PI3K-Akt signaling pathway, and fluid shear stress and atherosclerosis.

### 2.2. Proteomics Analysis Results

#### 2.2.1. iTRAQ Proteomics Analysis of the Liver

iTRAQ analysis of rat liver showed that a total of 5447 proteins were identified (Appendix A). Basic information statistics such as total spectra, peptides, and proteins were provided in Appendix A. Compared with the model group, a total of 598, 242, 130, and 159 DEPs were identified in the control, ASA, Eug, and AEE groups, respectively (Appendix A). In addition, the overlap of the DEPs presented by the Venn diagram (Appendix A) indicated that there were 419, 47, 65, and 77 DEPs in the model, ASA, Eug, and AEE groups, respectively. Protein quantitative and variance analysis list in Appendix A.

#### 2.2.2. Molecular Function and Pathway Analysis

To better understand the biological function of DEPs, GO and KEGG enrichment analyses were used. The ranking of the top 20 terms of the GO enrichment analysis is shown in Appendix A. In the category of biological process (BP), the DEPs from the model were mainly involved in the following processes: the prostanoid metabolic process, blood coagulation, fibrin clot formation, and the fatty acid biosynthetic process. The DEPs from the ASA group were related to oxygen transport and the regulation of the cholesterol metabolic process. Cellular response to hypoxia, cellular response to decreased oxygen levels, and response to hypoxia were mainly enriched for the DEPs from the Eug group. DEPs in the AEE group were mainly associated with positive regulation of vasoconstriction.

The results from KEGG pathway enrichment analysis are shown in Table 1 and Figure 2A–D. It was found that the most enriched pathway of the DEPs in the model was complement and coagulation cascades with protein counts of 26. Meanwhile, peroxisome, arachidonic acid metabolism, mineral absorption, and the PPAR and MAPK signaling pathways were found. KEGG pathways, such as the FoxO signaling pathway and the cell cycle, were found in the DEPs in the ASA group. Similarly, the pathways of phenylalanine, tyrosine, and tryptophan biosynthesis, vitamin digestion and absorption, and complement and coagulation cascades were enriched in the DEPs in the Eug group. Notably, platelet activation and complement and coagulation cascades were the significant pathways involved in the DEPs in the AEE group. Clearly, these pathways have a strong relationship with thrombosis.

Interaction networks of the DEPs were analyzed by the STRING database. Figure 3 showed that PPI results were in accordance with the GO and KEGG pathway analysis. It was found that 22 proteins involved in complement and coagulation cascades play pivotal roles in the PPI network of the DEPs from the model group (Figure 3A). Meanwhile, these DEPs were related to peroxisome, fatty acid metabolism, and hemostasis. As for the DEPs in the Eug and ASA groups, the PPI network indicated that a number of DEPs had a connection with retinol metabolism and complement and coagulation cascades (Figure 3B,C). Intriguingly, DEPs from the AEE group, such as fibrinogen gamma chain (Fgg), fibrinogen alpha chain (Fga), fibrinogen beta chain (Fgb), complement C6 (C6), complement C1s (C1s), G protein subunit alpha i1 (Gnai1), G protein subunit gamma 12 (Gng12), diacylglycerol kinase theta (Dgkq), protein tyrosine phosphatase non-receptor type 1 (Ptpn1), transferrin (Tf), lithogenic gene 4 (Lith4), syndecan 4 (Sdc4), syndecan 1 (Sdc1), and EH-domain containing 2 (Ehd2), were associated with complement and coagulation cascades, hemostasis, and platelet activation, signaling, and aggregation (Figure 3D).

#### 2.2.3. PRM Validation

In order to validate the proteomics data obtained from iTRAQ, six proteins, including Fga, Fgg, Fgb, orosomucoid 1 (Orm1), hemopexin (Hpx), and kininogen-2 (Kng2), were selected for PRM analysis. The fold change of these proteins in the PRM results was consistent with iTRAQ analysis, indicating the credibility of proteomics data (Table 2). Compared with the control, six proteins were upregulated in the model group, while ASA, Eug, and AEE reversed the tendency of upregulation of these proteins in varying degrees.

### 2.3. Metabonomics Analysis Results

#### 2.3.1. Multivariate Statistical Analysis and Metabolites Identification

Metabonomic data of liver tissue in rats using UPLC-Q-TOF/MS was provided in Appendix A. Based on the PCA score plots (Figure 4), three QC samples in ESI+ and ESI− were tightly clustered, suggesting the stability and reproducibility of the method. Meanwhile, a clear separation was found, which indicated the metabonomic profile of the liver sample differed among the control, the model, and the drug treatment groups. Metabonomic data in File S2.

The supervised pattern recognition method OPLS-DA was employed to determine the different metabolites. In the score plots of the OPLS-DA models (Figure 5), the liver samples in the control, ASA, AEE, and Eug groups were clearly separated from those in the model. With the VIP > 1 and the *p*-value < 0.05, 55 significantly different metabolites were found and identified. Compared with the control group, 17 metabolites were upregulated, and 38 were downregulated in the model group. Notably, AEE, ASA, and Eug modulated the κ-carrageenan-induced abnormal metabolites to the normal levels to some degree (Table 3). In order to intuitively display the relationships and find the differences among the groups, the clustering heatmap was generated for the metabolites (Figure 6A), which showed relative intensities of the metabolites and the cluster of liver samples in different groups.

#### 2.3.2. Metabolic Pathway and Function Analysis

MetaboAnalyst was used for the pathway analysis and metabolite set enrichment analysis (MSEA). Figure 6B,C shows the results of pathway analysis and MSEA indicating the altered metabolites were mainly involved in the metabolic pathways such as phenylalanine, tyrosine and tryptophan biosynthesis, linoleic acid metabolism, arachidonic acid metabolism, histidine metabolism, valine, leucine and isoleucine biosynthesis, sphingolipid metabolism, tricarboxylic acid (TCA) cycle and glutathione metabolism.

### 2.4. Integrative Analysis of Transcriptomics and Proteomics Data

The Venn diagrams in Figure 7 display the correlations between mRNA and protein differences in different comparisons. The number of shared genes and proteins in control vs. model, eugenol vs. model, and AEE vs. model were 28, 3, and 3, respectively. The results of GO correlation analysis for integrated comparative transcriptomic and proteomic data are shown in Figure 8. The BP of DEGs and DEPs in the control vs. model comparison were mainly involved in processes such as biological regulation, regulation of biological processes, and negative regulation of biological processes (Figure 8A). It was found that 20 DEPs and 11 DEGs in the ASA vs. model comparison were related to lipid metabolic processes (Figure 8B). The main biological processes of DEPs and DEGs enriched in the eugenol vs. model comparison were anatomical structure development, multicellular organism development, system development, and cellular response to chemical stimulus (Figure 8C). The majority of the DEGs and DEPs in the AEE vs. model comparison were related to macromolecule modification, protein modification process, cellular protein modification process, and regulation of cellular protein metabolic process (Figure 8D). Obviously, the enriched BPs of DEPs and DEGs in each group were different.

## 3. Discussion

The preventive effects of AEE on thrombosis were confirmed in our previous studies [21,22], indicating that AEE may be a potential chemotherapeutic agent for thrombosis treatment. In this study, transcriptomic, proteomic, and metabolomic profiles of liver tissue were investigated to systematically explore the potential mechanism of AEE in rats with thrombosis induced by carrageenan. Changes and related biological pathways of different genes, proteins, and metabolites from the livers in ASA-, eugenol-, and AEE-treated rats were characterized. From the perspective of omics, this study could provide insights into the mechanism of AEE and may help to understand the difference between aspirin, eugenol, and AEE in thrombosis treatment.

As an inflammation-inducing reagent, carrageenan can promote thrombosis formation through the activation of inflammatory molecule expression [19]. In this study, KEGG pathway enrichment analysis found that inflammatory signaling pathways such as the JAK-STAT signaling pathway, the NF-κB signaling pathway, and the TNF signaling pathway were enriched in the DEGs from the control vs. model, which indicated the increased levels of inflammation in the liver after carrageenan injection. The complement and coagulation cascades are key mediators in innate immunity and blood coagulation, which play essential roles in the thrombosis progression [25]. It was found that there were significant alterations in the expression of 10 genes (Fgb, C4a, C1s, C9, Fgg, Fga, C4bpa, Kng1, Serping1, and C4bpb) and 26 proteins related to complement and coagulation cascades. Therefore, the integrative transcriptomics and proteomics data suggested the disorders in the complement and coagulation cascades pathway were involved in the pathogenesis of thrombosis induced by carrageenan. It was found that arachidonic acid metabolism, including eight DEPs from the control vs. model, was enriched. Many studies proved that arachidonic acid and its metabolites, particularly prostaglandins (PGs), had proinflammatory action in many diseases [26]. Meanwhile, the metabolomics results indicated that the arachidonic acid level in the model group was 1.34-fold higher than that in the control. These results suggested that arachidonic acid metabolism was promoted in the rat after carrageenan injection. Linoleic acid had antithrombotic effects in thrombosis mice induced by collagen and adrenaline, which could enhance the survival rate and prolong the hemorrhage and coagulation time [27]. In this study, linoleic acid and some amino acids such as histidine, leucine, isoleucine, lysine, and phenylalanine were significantly reduced in the model group than those in the control, indicating the disorders in the metabolism of linoleic acid and amino acids might be responsible for carrageenan-induced thrombosis.

Ninety-four DEGs were found in the ASA group compared to the model group. These DEGs were mainly associated with the metabolism of linoleic acid, arachidonic acid, and amino acids. Linoleic acid is a type of unsaturated fatty acid and is also widely recognized as an essential fatty acid [28]. Linoleic acid can decrease blood cholesterol and prevent atherosclerosis [29]. Cholesterol needs to bind with linoleic acid for proper functioning and metabolism in the body [30]. Linoleic acid metabolism was interfered in inflammatory mice with reduced serum linoleic acid levels [31]. ASA upregulated linoleic acid levels [23]. This study’s results are consistent with them. Arachidonic acid is associated with many physiological and pathological processes, especially inflammatory responses. The plasma metabolomics results showed that the arachidonic acid content was higher in the ASA group compared to the carrageenan group [23]. In contrast, this study showed that arachidonic acid levels were higher in the model group than in the control group, while the ASA group could reverse this phenomenon. The above-mentioned phenomena may be due to the reduced synthesis of arachidonic acid in the liver, but it accumulates in the blood due to the inhibition of cyclooxygenase activity by ASA, which prevents the conversion of arachidonic acid to prostaglandins. Amino acid catabolism releases energy. This study showed that some amino acid levels were significantly lower in the model group than in the control group, while the ASA group could upregulate them. The results of metabolomics confirm this result. These data indicated that ASA has hypolipidemic, anti-inflammatory, and promoting energy metabolism effects. Compared with the model group, 242 DEPs were identified in the ASA group. These DEPs were mainly associated with the FoxO signaling pathway, with the regulation of oxygen transport and cholesterol metabolic processes. Carrageenan-induced thrombus affects blood supply, causing hypoxia, and the organism may improve this stress through increasing proteins related to oxygen transport [19]. Compared to the model group, ASA may improve the hypoxic environment by inhibiting the formation of carrageenan-induced thrombus. Cholesterol metabolic process regulation was significantly enriched in the biological process GO analysis, which involves two proteins, B1WBY7 (ER lipid raft-associated (1) and P80299 (Bifunctional epoxide hydrolase (2). B1WBY7 is a protein in the endoplasmic reticulum associated with the negative regulation of lipid biosynthesis [32], and the P80299 protein has both epoxide hydrolase activity and phosphorylation activity [33]. Compared with the model group, ASA significantly increased the levels of B1WBY7 and P80299 in the liver, which not only inhibited cholesterol biosynthesis but also accelerated the process of cholesterol catabolism. ASA has a certain regulatory effect on dyslipidemia and could reduce serum triglycerides, cholesterol, and other indexes [34]. The FoXo signaling pathway is a metabolic pathway closely related to erythrocyte signaling, vascular smooth muscle cell apoptosis, and value addition [35]. This indicated that ASA may affect metabolic pathways related to erythrocytes or vascular smooth muscle cells. These data indicated that ASA may play a thromboprophylaxis role through the regulation of cholesterol metabolism, anti-inflammation, and promotion of energy metabolism.

Two hundred and forty-two DEGs were identified in the Eug group compared to the model group. These DEGs were mainly associated with the mTOR signaling pathway, the Toll-like receptor signaling pathway, and the IL-17 signaling pathway. MTOR is a protein kinase that regulates lipid metabolism, protein synthesis, and energy metabolism through downstream effectors. The MTOR signaling pathway is activated to accelerate protein synthesis, promote liposynthesis and adipocyte differentiation, and influence adipocyte stability [36]. This indicated that Eug may affect metabolic pathways related to lipid metabolism, protein synthesis, and energy metabolism. Toll-like receptors (TLRs) are pattern recognition receptors that mediate the recognition of and response to foreign pathogens, and they play a key role in inflammation and immune cell regulation [37]. IL-17 is a key pro-inflammatory cytokine produced mainly by Th17 cells and is involved in multiple inflammatory responses [38]. Huangqi Chifeng decoction downregulated the expression of factors related to the IL-17 signaling pathway in the immunoglobulin A nephropathy model of rats, improved blood stasis, and reduced pathological kidney injury in rats [39]. This indicated that Eug may affect metabolic pathways associated with the inflammatory response. Compared with the model group, 130 DEPs were identified in the Eug group. These DEPs were mainly associated with the complement and coagulation cascade pathways, cellular responses to hypoxia, cellular responses to low oxygen levels, and responses to hypoxia. In GO analyses of biological processes, there was a significant enrichment of terms related to hypoxic stress, which contained the proteins Q498D8 (Ring-box 1), P06762 (Heme oxygenase 1), A0A0G2JV72 (Ubiquitin-conjugating enzyme E2 D3), Q6P136 (Hyou1 protein), P29975 (Aquaporin-1), and Q63400 (Claudin-3). This showed that Eug has a role in hypoxic stress-related proteins. KEGG-enriched results showed significant enrichment of complement and coagulation cascade reactions, and Eug significantly reduced the expression level of related proteins, as well as some effects on vitamin metabolism and phenylalanine, tyrosine, and tryptophan biosynthesis processes.

One hundred and thirty-two DEGs were identified in the AEE group compared with the model group. These DEGs were mainly associated with the IL-17 signaling pathway, the MAPK signaling pathway, the PI3K-Akt signaling pathway, fluid shear stress, and atherosclerosis. MAPK is considered to be a regulator of numerous genes, and it plays a regulatory role in protein expression [40]. Inflammatory cytokines, pathogens, and oxidative stress can activate MAPK. MAPK is one of the indicators of inflammation, which is closely related to inflammation and is often a part of various pathways of inflammation [41]. The PI3K/Akt signaling pathway acts as a bridge between extracellular signals and cellular responses, enabling the downward transmission of signaling molecules, thus playing a role in apoptosis and glycolipid metabolism [42]. This indicated that AEE may affect metabolic pathways related to inflammatory response and energy metabolism. Compared with the model group, 159 DEPs were identified in the AEE group. These DEPs were mainly associated with platelet activation and complement and coagulation cascade reactions. GO molecular function enrichment results showed that the different proteins between AEE and the model group were mainly involved in the phosphorylation regulation process, indicating that AEE may have some influence on the phosphorylation modification of proteins in the liver. KEGG pathway-enriched results showed that the complement and coagulation cascade response involved seven differential proteins. The results showed that AEE significantly reduced the expression of these seven proteins in the liver, which implied a significant negative regulatory effect of AEE on the metabolic pathway of complement and coagulation cascade reaction. It indicated that AEE may inhibit the formation and development of thrombus through the negative regulation of this pathway. The platelet activation pathway was also significantly enriched in KEGG analysis, which involved six proteins, three of which were the same as those involved in the complement and coagulation cascade reactions. This result indicated that two metabolic pathways were closely linked, and AEE may inhibit platelet aggregation by negatively regulating the complement and coagulation cascade. These data indicated that AEE could play a thromboprophylaxis role by regulating platelet activation and complement and coagulation cascade reactions.

Six proteins were screened for PRM validation, including P06399 (Fga), P14480 (Fgb), and P02680 (Fgg), which are the α, β, and γ chains of fibrinogen, respectively, and the other three were P02764 (Orm1), P20059 (Hpx), and Q5PQU1 (Kng2). Fibrinogen, also known as coagulation factor I, is a protein with coagulation function synthesized by the liver, which can increase blood viscosity and promote platelet aggregation, which is very important in cardiovascular diseases [43]. PRM results showed that all three types of fibrinogens were significantly increased in the liver of the model group compared to the control group, which indicated that carrageenan could activate the coagulation pathway by increasing the expression of fibrinogen, thereby inducing thrombosis. AEE, ASA, and Eug were able to significantly reverse carrageenan-induced fibrinogen increases in the liver. This indicated an ameliorative effect of AEE, ASA, and Eug on carrageenan-induced thrombosis. Orm1 is one of the serum acute phase proteins, also known as mucin-like protein, which is mainly synthesized by the liver. Its expression is significantly increased during the occurrence of inflammation, tissue damage, and infection, and it is regarded as one of the indicators of the occurrence of inflammation [44]. PRM results showed that Orm1 levels in the model group were 3.84 times higher than those in the control group, which indicated that carrageenan induced acute systemic inflammation in rats. AEE and Eug decreased the hepatic expression levels of Orm1, while ASA did not have a significant effect on this protein. Hpx is the protein in plasma with the strongest affinity for hemoglobin and is mainly synthesized in the liver, and stress responses can lead to increased levels of its expression. Inflammatory factors such as interleukins and tumor necrosis factor can induce the expression of this protein [45]. Similar to the Orm1 results, carrageenan increased the expression level of Hpx in the liver, and ASA, AEE, and Eug all decreased the hepatic expression level of this protein. Kng2 is an important type of active polypeptide in organisms, and its expression level was significantly increased after inflammation induction [46]. Compared with the control group, carrageenan significantly increased the level of Kng2 in the liver of rats in the model group. AEE and Eug significantly reduced the hepatic expression of Kng2 and contributed to inhibiting inflammation and coagulation reaction, while ASA did not have a significant effect on this protein.

Metabolomics results showed that the expression of 17 metabolites was upregulated, and 38 were downregulated in the model group compared to the control group. In contrast, AEE, ASA, and Eug could reverse the expression of these metabolites to different degrees and make them back to normal levels. These metabolites are mainly involved in metabolic pathways such as phenylalanine, tyrosine, and tryptophan biosynthesis; linoleic acid metabolism; arachidonic acid metabolism; histidine metabolism; valine, leucine, and isoleucine biosynthesis; sphingolipid metabolism; the citric acid cycle; and glutathione metabolism. The liver is the main organ involved in amino acid metabolism and is primarily responsible for maintaining the body’s balance of amino acids [47]. Compared with the control group, the levels of amino acids were significantly reduced in the liver of the model group. The levels of amino acids were able to increase significantly by treatment with AEE, ASA, and Eug, which indicated that they were able to normalize the abnormal liver amino acid metabolism. Therefore, it was speculated that the thromboprophylaxis effects of AEE, ASA, and Eug were associated with the promotion of amino acid metabolism. Methionine plays an important role in lipid metabolism, oxidative stress, and bile metabolism [48]. Valine and leucine could produce succinyl CoA through a degradation reaction and thus enter the tricarboxylic acid cycle [49]. AEE, ASA, and Eug lead to an increase in liver levels of methionine, valine, and leucine, which could help improve lipid and bile metabolism, reduce oxidative stress, and repair energy metabolism damage. These changes are all beneficial in the treatment of thrombosis. Malic acid is an intermediate metabolite of the TCA. The reduced levels of malate in the model group implied inhibition of energy production, and AEE, ASA, and Eug treatment corrected the malate abnormality, which indicated that AEE, ASA, and Eug had a modulating effect on energy metabolism dysfunction. These data indicated that the thromboprophylaxis effects of AEE, ASA, and Eug were associated with improved lipid and bile metabolism, reduced oxidative stress, and repair of energy metabolism damage. This study also has some limitations; for example, the experimental results were not systematically validated, and a single animal model and cross-validation of multiple animal models were not performed.

## 4. Materials and Methods

### 4.1. Reagents and Chemicals

Acetonitrile (MS-grade) and methanol (MS-grade) were purchased from Thermo Fisher Scientific Corporation (Waltham, MA, USA). Carboxymethylcellulose sodium (CMC-Na), ASA, and eugenol were obtained from Aladdin Industrial Corporation (Shanghai, China). AEE was synthesized and purified at the Lanzhou Institute of Husbandry and Pharmaceutical Sciences of CAAS. AEE, ASA, and Eug were ground, and then their suspensions were prepared in 0.5% CMC-Na. Deionized water (18 MΩ) was prepared with a Direct-Q^®^3 system (Millipore, Billerica, MA, USA). MS-grade formic acid was purchased from TCI (Shanghai, China), and κ-carrageenan was provided by Sigma-Aldrich (St. Louis, MO, USA). iTRAQ reagent multi-plex kit was supplied by Applied Biosystems (Foster City, CA, USA).

### 4.2. Animals and Sampling

Male Wistar rats (*n* = 45) weighing 230–250 g, supplied by Lanzhou Veterinary Research Institute (Lanzhou, China), were housed in standard animal conditions and free access to standard diet and water. All the rats were randomly separated into 5 groups as follows: control, model, AEE, ASA, and Eug groups (*n* = 9). In drug treatment groups, suspensions of AEE (36 mg/kg), ASA (20 mg/kg), and Eug (18 mg/kg) were administered intragastrically to the rats. For the comparison of the results, the molar quantities of AEE, ASA, and Eug were designed to equal 0.11 mmol. Meanwhile, the rats in the control and model groups were treated with an equal volume of solvent (0.5% CMC-Na). With one-week drug administration, the rats were intraperitoneally injected with 20 mg/kg κ-carrageenan to induce thrombosis. After 48 h, all the rats were anesthetized with sodium pentobarbital (20 mg/kg), and then livers were immediately removed. Liver samples from three individual rats in the same group were pooled as a biological sample, resulting in three biological replicates for transcriptomics and proteomics analysis. The animal experiment was approved by the Institutional Animal Care and Use Committee of the Lanzhou Institute of Husbandry and Pharmaceutical Science of the Chinese Academy of Agricultural Science (Approval No. NKMYD202211). Animal welfare and experimental procedures were performed strictly in accordance with the Guidelines for the Care and Use of Laboratory Animals issued by the United States National Institutes of Health.

### 4.3. Transcriptomics for RNA Isolation, Library Preparation, Sequencing, and Enrichment Analysis

Total RNA in liver tissue samples was isolated using TRIzol reagent (Life Technologies, Carlsbad, CA, USA) and further purified using RNeasy Mini Kit and RNase-Free DNase Set (QIAGEN, GmBH, Hilden, Germany). RNA degradation and contamination were monitored on 1% agarose gels, and the purity was checked using the NanoPhotometer spectrophotometer (IMPLEN, Westlake Village, CA, USA). RNA integrity was assessed using the RNA Nano 6000 Assay Kit (Agilent Technologies, Santa Clara, CA, USA). Sequencing libraries were generated using NEBNext Ultra RNA Library Prep Kit for Illumina (NEB, Ipswich, MA, USA) following the manufacturer’s recommendations. PCR was performed to enrich and amplify the cDNA fragments, and the PCR products were purified (AMPure XP system). Then, library quality was assessed on the Agilent Bioanalyzer 2100 system. The resultant libraries were sequenced on an Illumina Hiseq 2500 platform, and paired-end reads were generated.

Quality control of the raw reads was performed by using FastQC software v0.10.1 (https://github.com/s-andrews/FastQC/), and the remaining clean reads were used for further analysis. STAR software (version 2.4.1a) was used to align the clean reads to the rat genome. Fragments per kilobase of transcript per million fragments mapped (FPKM) method was used to normalize gene expression. Differentially expressed genes (DEGs) were selected using DESeq2 with strict criteria of fold change > 2 and *p*-value < 0.05 (FDR < 0.05). The *p*-value was adjusted by the false discovery rate (FDR < 0.05). Venn plot generated by online software Venny (version 2.1) was used to visualize the overlap of DEGs. Functional annotations of the DEGs were performed using the KEGG database. The enrichment analysis of DEGs was implemented by the clusterProfiler R package (version 3.10.1). KEGG pathways with *p*-values less than 0.05 were considered significant.

### 4.4. ITRAQ Proteomics

#### 4.4.1. Protein Exaction and Labeling

Liver sample preparation, protein extraction, protein digestion, and peptide tagging were carried out as previously described. Briefly, liver tissues were ground into powders in liquid nitrogen and dissolved in SDT buffer (4% SDS, 100 mM Tris-HCl, 1 mM DTT, pH 7.6). The lysate was homogenized and sonicated, and then boiled for 15 min. Following the centrifugation at 14,000× *g* for 40 min, the supernatant was collected and filtered with 0.22 µm filters. The protein was quantified by BCA assay. Next, the pooled samples were digested with the filter-aided sample preparation (FASP) method, and then the peptide mixture was labeled with the 8-plex iTRAQ reagent (Applied Biosystems Foster, CA, USA). Labeled peptides were fractionated by strong cation-exchange (SCX) chromatography using the AKTA Purifier system (GE Healthcare, Chicago, IL, USA). Finally, collected fractions were combined and desalted on C18 Cartridges (Empore™ SPE Cartridges C18 (standard density), bed I.D. 7 mm, volume 3 mL, Sigma-Aldrich, St. Louis, MO, USA).

#### 4.4.2. LC-MS/MS Analysis of Proteins

The peptide mixture was loaded onto the Thermo EASY-nLC System equipped with Acclaim PepMap100 trap column (100 μm × 2 cm, nanoViper C18, Thermo Fisher Scientific, Shanghai, China). Peptides were separated on analytical EASY column (75 μm × 10 cm, 3 μm, Thermo Scientific) over 60 min at a flow rate of 300 nL/min consisting of buffer A (0.1% formic acid) and buffer B (84% *v*/*v* acetonitrile and 0.1% *v*/*v* formic acid). The liquid-phase linear-gradient program was as follows: 0–35% buffer B for 50 min, 35–100% buffer B for 5 min, and hold in 100% buffer B for 5 min.

MS data were acquired using Q Exactive MS (Thermo Scientific) in the positive ion mode over 300–1800 *m*/*z* at a resolution of 70,000 at *m*/*z* 200. The automatic gain control (AGC) target was set to 3 × 10^6^, and the maximum inject time to 10 ms. Precursor ions for higher-energy collisional dissociation (HCD) fragmentation were dynamically selected according to a data-dependent top 10 method. Values for MS/MS analysis were set as follows: resolution for HCD spectra was 17,500 at *m*/*z* 200, isolation width was 2 *m*/*z*, normalized collision energy was 30 eV, dynamic exclusion duration was 40 s, and underfill ratio was defined as 0.1%.

#### 4.4.3. Protein Identification and Bioinformatics Analysis

Proteins were identified using the MASCOT search engine (version 2.2, Matrix Science, London, UK) with Proteome Discoverer 1.4 (Thermo Electron, San Jose, CA, USA). The parameters for protein identification were as follows: enzyme was set as trypsin, max missed cleavages were 2, peptide mass tolerance was set at ±20 ppm, fragment mass tolerance was 0.1 Da, and fixed modification was carbamidomethyl cysteine. In order to minimize the false positive results, the cutoff of FDR less than 0.01 was applied in the protein identification. Proteome Discoverer 1.4 software was used to determine the differentially expressed proteins (DEPs). Compared with the model, the proteins with a *p*-value less than 0.05 and higher than 1.2-fold changes (or lower than 0.83) were considered DEPs for further analysis.

Gene Ontology (GO) annotations of the proteins were performed using the Blast2GO program (version 3.3.5). The corresponding KEGG pathways of the proteins were also extracted and mapped in the KEGG database (http://geneontology.org/, accessed 23 November 2023). GO enrichment and KEGG pathway enrichment analysis were applied based on Fisher’s exact test. Functional protein–protein interaction (PPI) networks were analyzed by using STRING (http://string-db.org/, accessed 28 November 2023) and Cytoscape software (version 3.2.1).

#### 4.4.4. Verification Analysis by Parallel Reaction Monitoring

Candidate proteins, including Fga, Fgg, Fgb, Orm1, Hpx, and Kng2, were selected for verification by parallel reaction monitoring (PRM) analysis. Raw data of PRM analysis was processed in Skyline (version 3.5.0, MacCoss Lab., University of Washington, Washington, DC, USA).

### 4.5. Metabonomics Analysis

#### 4.5.1. Metabonomics Analysis Platform

Liver sample preparation was performed as previously described. Briefly, a 200 mg liver sample was added to 2 mL methanol/water (4:1, *v*/*v*) for homogenization, followed by 1 min of vortex mixing and 8 min of ultrasonic extraction. After standing (10 min on ice) and centrifugation (15,000 rpm, 10 min, 4 °C), 1.6 mL of the supernatant was transformed and evaporated to dry. The residue was reconstructed in 200 μL methanol/water (4:1, *v*/*v*) for metabonomics analysis. In addition, the pooled quality-control (QC) samples were prepared to monitor the MS data collection.

Chromatographic analysis was carried out using the 1290 UPLC system (Agilent Technologies, Santa Clara, CA, USA). An aliquot of 4 μL sample solution was injected into an Agilent ZORBAX SB-C18 column (2.1 × 150 mm, 1.8 μm). The column was maintained at 35 °C, and the flow rate was 0.25 mL/min. The mobile phase consisted of a linear-gradient system of (A) water with 0.1% formic acid and (B) acetonitrile with 0.1% formic acid: 0–2 min, 98% A; 2–11 min, 98–55% A; 11–15 min, 55–30% A; 15–22 min, 30–2% A; and 22–27 min, 2% A.

High-definition mass spectrometer Agilent 6530 Q-TOF (Agilent Technologies, Santa Clara, CA, USA) was used to perform the mass data acquisition in positive or negative electrospray ionization source (ESI+ or ESI−). The optimal conditions of MS analysis were as follows: MS data was collected in centroid mode from 50 to 1000 *m*/*z*, scan rate was 1 spectrum/second, desolvation gas rate was 10 L/min, gas temperature was 350 °C, nebulizer pressure was 45 psig, fragment voltage was 135 V, skimmer voltage was 65 V, and capillary voltages were 4.0 KV in positive mode and 3.5 KV in negative mode, respectively.

#### 4.5.2. Multivariate Statistical Analysis

The Mass Hunter Qualitative Analysis software (Version B.06.00, Agilent Technologies, Santa Clara, CA, USA) and XCMS were used for extraction alignment and integration of the peak intensities. Then, the data was normalized and introduced to SIMCA-P (V13.0, Umetrics AB, Umea, Sweden). Principal component analysis (PCA) and orthogonal partial least-squares discriminant analysis (OPLS-DA) were performed. Variable importance for projection (VIP) value and s-plots constructed from the OPLS were used to select the potential biomarkers. Metabolites with VIP > 1 and *p* < 0.05 were considered statistically significant.

#### 4.5.3. Identification and Pathway Analysis

Metabolite identification was performed by MS/MS analysis and database searching. Accurately measured *m*/*z* and MS/MS fragment ions were searched against the METLIN (https://metlin.scripps.edu, accessed 15 June 2023), Human Metabolome Database (HMDB) (http://www.hmdb.ca/, accessed 15 June 2023), MassBank (http://www.massbank.jp/, accessed 15 June 2023), mzCloud (https://www.mzcloud.org/, accessed 15 June 2023), and Lipid Maps (http://www.lipidmaps.org/, accessed 15 June 2023). Metabolic pathways and biochemical reactions of the metabolites were identified through the KEGG and HMDB databases. Pathway analysis and visualization were performed on MetaboAnalyst 4.0 (http://www.metaboanalyst.ca/, accessed 15 June 2023).

## 5. Conclusions

AEE could modulate the expression of multiple genes, protein abundance, and metabolite levels in rat liver in the κ-carrageenan-induced rat tail thrombosis model. A total of 132 differentially expressed genes, 159 differentially expressed proteins, and 55 differential metabolites were identified. AEE could reverse to some extent the changes in them caused by κ-carrageenan. Some DEPs associated with thrombosis were analyzed by PRM, including Fga, Fgg, Fgb, or m1, Hpx, and Kng2. Compared with the control group, the expression of all these proteins was upregulated in the model group. However, AEE reversed the upregulation of these proteins and also regulated the expression of metabolites, bringing them back to normal levels. Therefore, AEE had good preventive effects from thrombosis. As precursor drugs of AEE, both ASA and Eug had certain prophylactic effects on thrombosis, but the prophylactic effect of AEE was superior to that of ASA and Eug. AEE could prevent thrombosis by inhibiting platelet activation, reducing inflammation, and regulating amino acid and energy metabolism. We will perform multiple thrombus models to systematically validate the efficacy of AEE in preventing thrombosis and will explore the mechanism of AEE in preventing thrombosis from the perspective of platelet activation by whole transcriptomics and proteomics techniques for platelets.

## Figures and Tables

**Figure 1 ijms-25-02141-f001:**
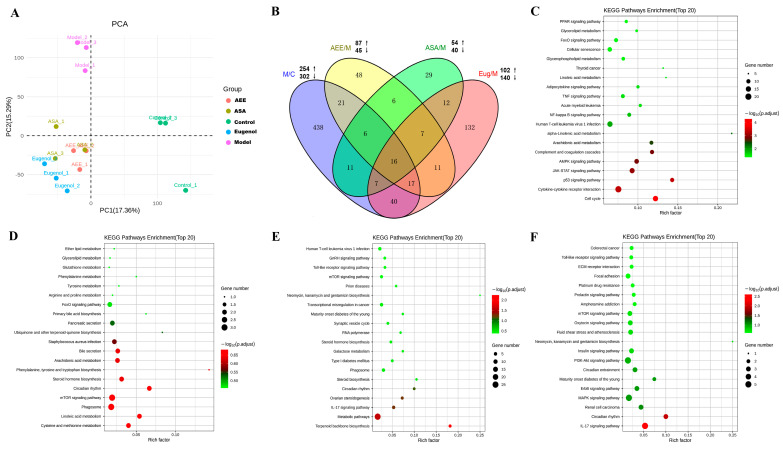
Profiling of DEGs in the liver tissues of rats from different groups. (**A**) Principal component analysis of transcriptomics data from the liver in different groups. (**B**) Venn diagram of DEGs in different group comparisons representing the unique and overlapping DEGs. M/C: model vs. control; AEE/M: AEE vs. model; ASA/M: aspirin vs. model; and Eug/M: eugenol vs. model. (**C**–**F**) The top 20 key KEGG pathways of the DEGs in different group comparisons: (**C**) M/C, (**D**) ASA/M, (**E**) Eug/M, and (**F**) AEE/M.

**Figure 2 ijms-25-02141-f002:**
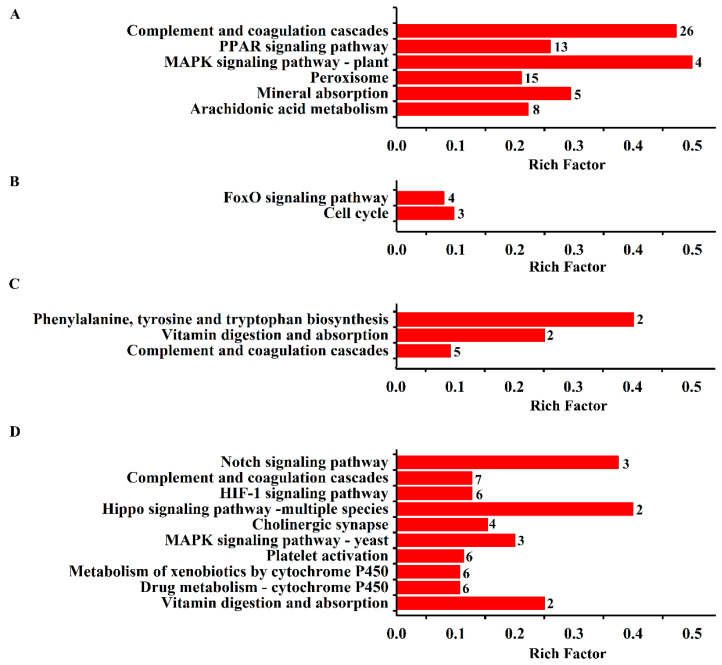
KEGG enrichment analysis of the DEPs in the liver tissue. The enriched biological processes are displayed, and the number of proteins involved is shown beside the bar. (**A**) DEPs from control vs. model; (**B**) DEPs from ASA vs. model; (**C**) DEPs from eugenol vs. model; and (**D**) DEPs from AEE vs. model.

**Figure 3 ijms-25-02141-f003:**
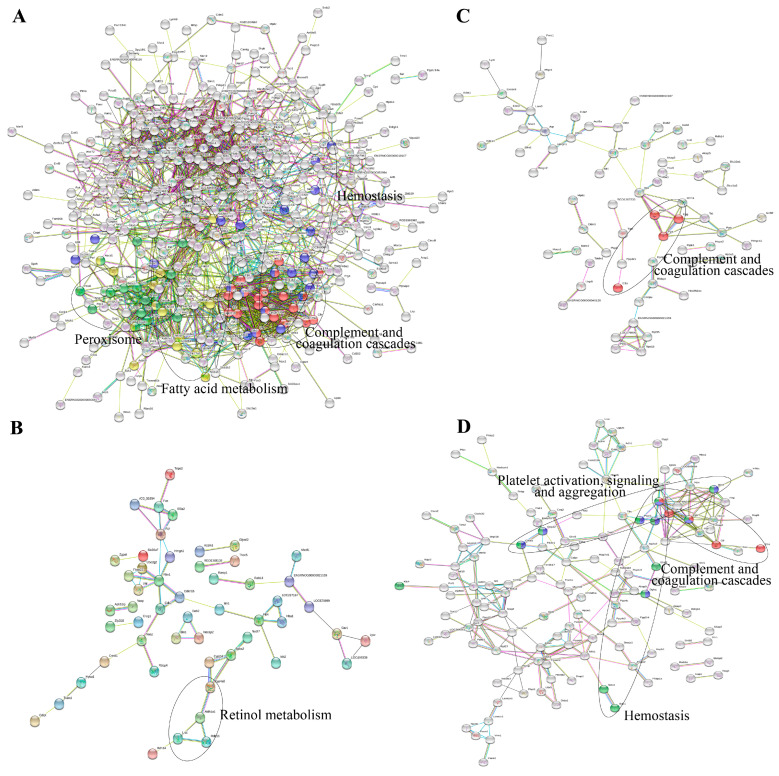
Interaction networks of the DEPs analyzed by the STRING database: (**A**) DEPs from control vs. model; (**B**) DEPs from ASA vs. model; (**C**) DEPs from eugenol vs. model; and (**D**) DEPs from AEE vs. model.

**Figure 4 ijms-25-02141-f004:**
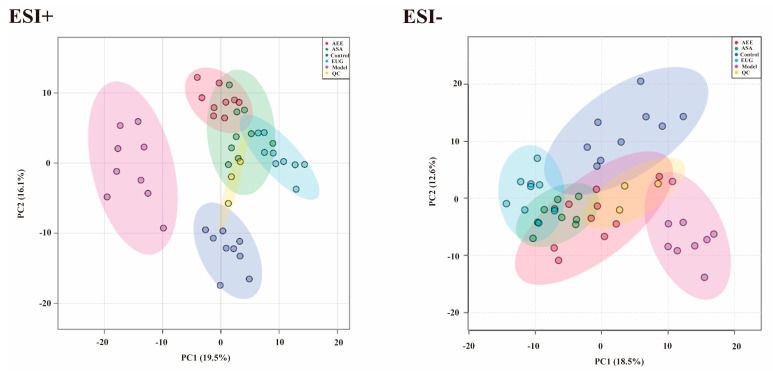
PCA score plots of the liver samples analyzed by the UPLC-Q-TOF/MS in ESI+ and ESI− modes. ESI+: electrospray ionization in positive ion mode; ESI−: electrospray ionization in negative ion mode. A separation trend was found among the five groups. Three QC samples were tightly clustered, indicating the stability of the system and the method.

**Figure 5 ijms-25-02141-f005:**
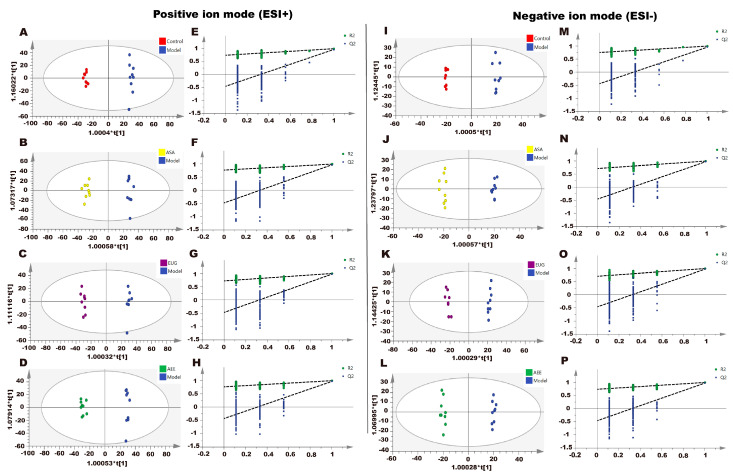
Effect of AEE on metabolomic profiles of liver in thrombosed rats. The OPLS-DA score plots of different groups in positive and negative modes. (**A**) Control group vs. model group, ESI+: R^2^X = 0.428, R^2^Y = 0.996, and Q^2^ = 0.963. (**B**) ASA group vs. model group, ESI+: R^2^X = 0.456, R^2^Y = 0.990, and Q^2^ = 0.965. (**C**) Eug group vs. model group, ESI+: R^2^X = 0.421, R^2^Y = 0.996, and Q^2^ = 0.972. (**D**) AEE group vs. model group, ESI+: R^2^X = 0.442, R^2^Y = 0.994, and Q^2^ = 0.966. (**E**–**H**) Permutation test of OPLS-DA models in ESI+. (**I**) Control group vs. model group, ESI−: R^2^X = 0.400, R^2^Y = 0.995, and Q^2^ = 0.952. (**J**) ASA group vs. model group, ESI−: R^2^X = 0.391, R^2^Y = 0.993, and Q^2^ = 0.967. (**K**) Eug group vs. model group, ESI−: R^2^X = 0.449, R^2^Y = 0.995, and Q^2^ = 0.972. (**L**) AEE group vs. model group, ESI−: R^2^X = 0.412, R^2^Y = 0.996, and Q^2^ = 0.970. (**M**–**P**) Permutation test of OPLS-DA models in ESI−.

**Figure 6 ijms-25-02141-f006:**
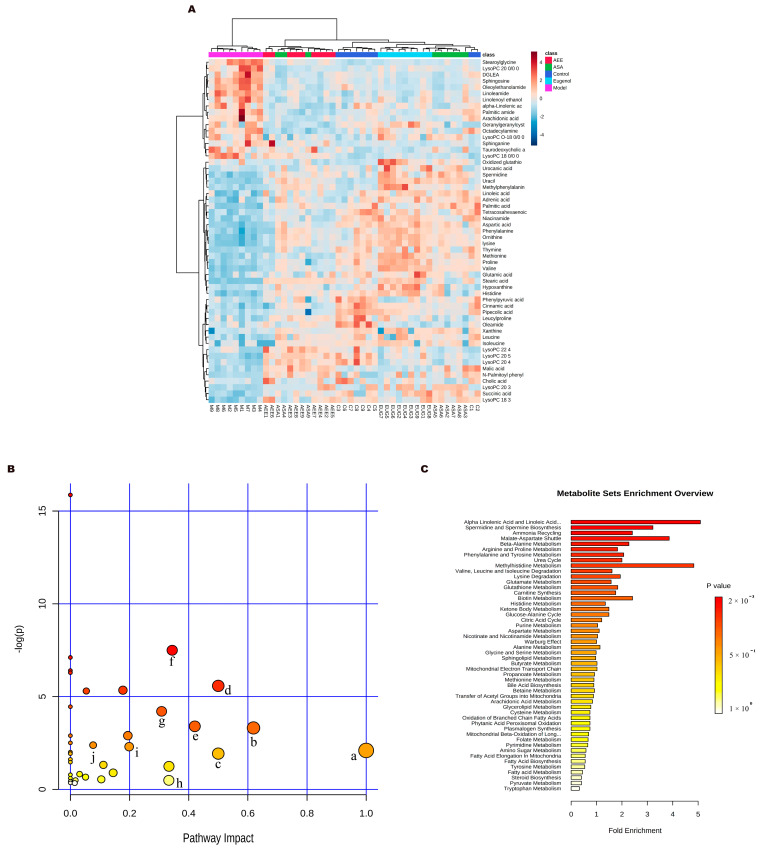
Metabonomics analysis results. (**A**) Heat map of metabolite clustering. (**B**) Metabolite pathway analysis: a: linoleic acid metabolism; b: phenylalanine metabolism; c: D−glutamine and D−glutamate metabolism; d: phenylalanine, tyrosine, and tryptophan biosynthesis; e: alanine, aspartate, and glutamate metabolism; f: histidine metabolism; g: arginine and proline metabolism; h: arachidonic acid metabolism; i: sphingolipid metabolism; j: citrate cycle. (**C**) Metabolite enrichment analysis.

**Figure 7 ijms-25-02141-f007:**
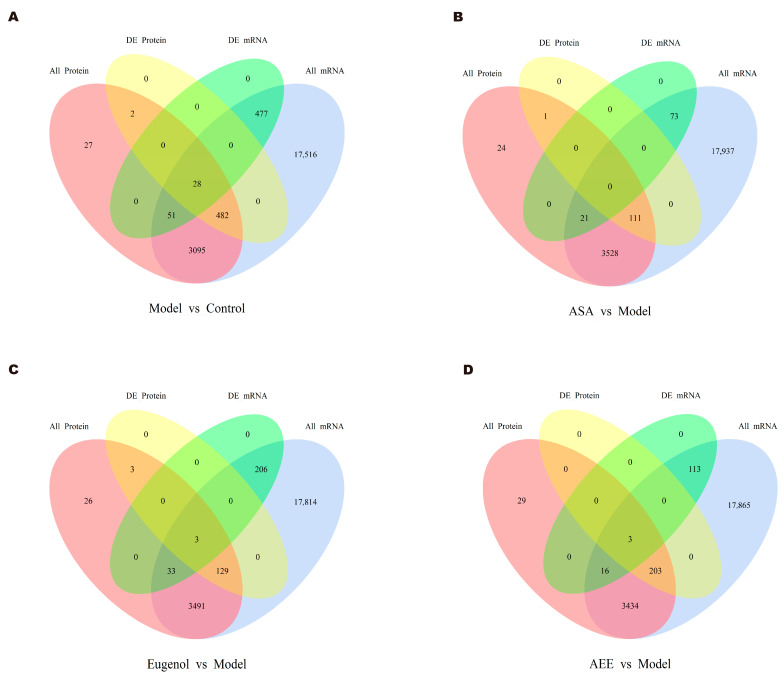
Venn diagrams for correlation analysis of mRNA and protein differences: (**A**) control group vs. model group; (**B**) ASA group vs. model group; (**C**) EUG group vs. model group; and (**D**) AEE group vs. model group.

**Figure 8 ijms-25-02141-f008:**
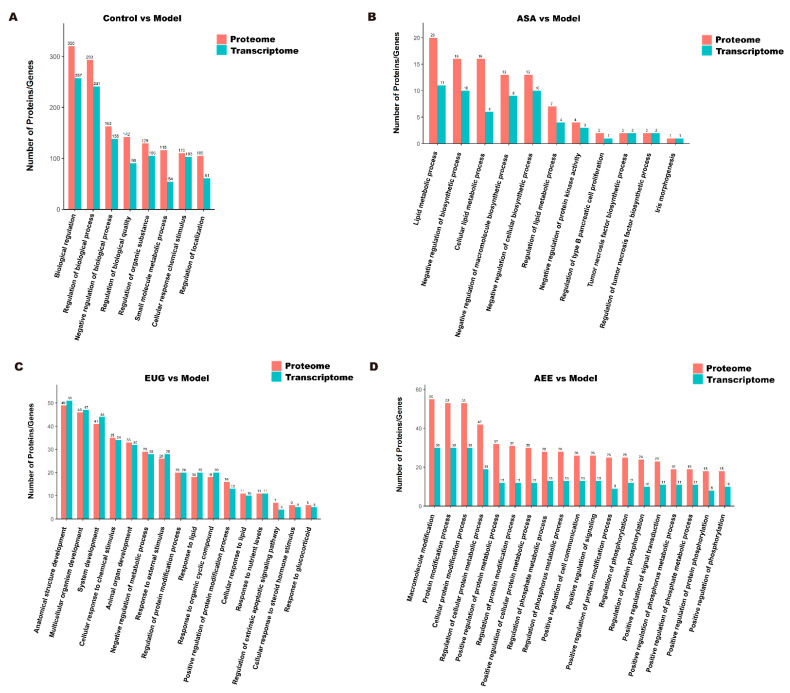
GO correlation analysis of transcriptomic and proteomic data: (**A**) control group vs. model group; (**B**) ASA group vs. model group; (**C**) EUG group vs. model group; and (**D**) AEE group vs. model group.

**Table 1 ijms-25-02141-t001:** KEGG pathway enrichment analysis of the DEPs.

Map ID	Pathway Name	Proteins Accession ID	Count	RF	*p*-Value
Control vs. Model					
map04610	Complement and coagulation cascades	P08932 P19999 Q3KR94 D4A1S0 Q62930 Q5PQU1 P06399 P02680 A0A096P6L9 P14480 Q6MG74 Q5M7T5 D3ZWD6 A0A0G2JY31 G3V836 Q9EQV9 F1M983 Q6MG73 Q6P734 B5DEH7 P08649 P16296 F1M7F7 M0R5R0 G3V843 P31394	26	0.47	0.0000
map03320	PPAR signaling pathway	P20817 P07379 P55051 G3V8J2 Q9ES38 P24464 P07896 P07308 F1LNW3 F1LQC1 P55053 D3ZJX6 A0A0G2K8Q1	13	0.26	0.0022
map04016	MAPK signaling pathway—plant	P19804 P62161 P04762 G3V816	4	0.50	0.0070
map04146	Peroxisome	P07895 P56574 F1LNG8 Q8CHM7 P04762 B0BNF9 P07896 F1LNW3 F1LQC1 D3ZDM7 D3ZZB2 Q5I0K1 P54777 D3ZHR2 O35078	15	0.21	0.0089
map04978	Mineral absorption	Q9WUC4 Q7M082 D3ZHV3 P12346 Q66HI5	5	0.29	0.0316
map00590	Arachidonic acid metabolism	O35543 P20817 P24464 P11510 Q5EB99 D3ZJX6 Q02759 B2GV28	8	0.22	0.0379
**AEE vs. Model**					
map04330	Notch signaling pathway	P97887 B3DM90 Q2LC86	3	0.38	0.0041
map04610	Complement and coagulation cascades	F1M7F7 Q5PQU1 P06399 P02680 A0A0G2JY31 P14480 D4A1S0	7	0.13	0.0102
map04723	Retrograde endocannabinoid signaling	G3V6P8 Q45QM4 Q45QM8 Q5XI64	4	0.19	0.0125
map04066	HIF-1 signaling pathway	Q498D8 O08769 P12346 G3V679 E9PTN6 M0RC47	6	0.13	0.0168
map04392	Hippo signaling pathway—multiple species	P35465 R9PXS9	2	0.40	0.0179
map04725	Cholinergic synapse	G3V6P8 Q45QM4 Q45QM8 M0RC47	4	0.15	0.0263
map04011	MAPK signaling pathway—yeast	P35465 D3ZAP9 Q4V886	3	0.20	0.0265
map04611	Platelet activation	Q45QM4 Q45QM8 M0RC47 P06399 P02680 P14480	6	0.11	0.0289
map00980	Metabolism of xenobiotics by CYP450	A1L128 P04903 O35543 P06757 A9EEP5 F1LM22	6	0.11	0.0366
map00982	Drug metabolism—CYP450	A1L128 P04903 O35543 P06757 A9EEP5 F1LM22	6	0.11	0.0366
map04977	Vitamin digestion and absorption	Q5FVF9 Q9JI61	2	0.25	0.0461
**ASA vs. Model**					
map04068	FoxO signaling pathway	Q811R1 O08769 Q63699 Q9Z2X5	4	0.08	0.0306
map04110	Cell cycle	Q498D8 O08769 Q63699	3	0.10	0.0367
**Eug vs. Model**					
map00400	Phenylalanine, tyrosine, and tryptophan biosynthesis	P04176 P04694	2	0.40	0.0080
map04977	Vitamin digestion and absorption	Q5FVF9 Q9JI61	2	0.25	0.0211
map04610	Complement and coagulation cascades	F7FEU1 D4A1S0 Q5PQU1 P06399 P02680	5	0.09	0.0214

Control vs. Model, control group compared with model group; AEE vs. Model, AEE group compared with model group; ASA vs. Model, ASA group compared with model group; Eug vs. Model, Eug group compared with model group.

**Table 2 ijms-25-02141-t002:** PRM validation of proteins.

Protein ID	Gene	Description	M/C		ASA/M		Eug/M		AEE/M	
			iTRAQ	PRM	iTRAQ	PRM	iTRAQ	PRM	iTRAQ	PRM
P06399	Fga	Fibrinogen alpha chain	1.71 **	1.17	0.90	0.96	0.81 *	0.47 *	0.77 *	0.81
P02680	Fgg	Fibrinogen gamma chain	1.69 **	1.33	0.88	0.85	0.82 *	0.43 **	0.76 *	0.79
P14480	Fgb	Fibrinogen beta chain	1.58 **	1.33	0.90	0.83	0.86	0.37 **	0.81 *	0.76
P02764	Orm1	Alpha-1-acid glycoprotein	2.52 **	3.84 **	0.96	1.16	0.81 *	0.80	0.67 **	0.70
P20059	Hpx	Hemopexin	1.62 **	1.32	0.83 *	0.94	0.82 *	0.79	0.80 *	0.80
Q5PQU1	Kng2	Kininogen 2	2.08 **	5.72 **	1.00	1.11	0.81 **	0.68	0.83 *	0.82

* *p* < 0.05, ** *p* < 0.01. M/C: model vs. control; AEE/M: AEE vs. model; ASA/M: aspirin vs. model; Eug/M: eugenol vs. model.

**Table 3 ijms-25-02141-t003:** Statistics of different metabolites in rat liver.

No.	Metabolites	Formula	Ion Mode	*m*/*z*	RT (min)	Fold Change			
						Mod/C	ASA/Mod	Eug/Mod	AEE/Mod
1	Cinnamic acid	C_9_H_8_O_2_	ESI−	147.0465	5.36	0.56 **	1.30 **	1.42 **	1.13
2	Xanthine	C_5_H_4_N_4_O_2_	ESI−	151.0275	2.39	0.64 **	1.47 *	1.40	1.26 *
3	Cholic acid	C_24_H_40_O_5_	ESI−	407.2843	14.47	0.37 **	0.98	1.47	2.18
4	Palmitic acid	C_16_H_32_O_2_	ESI−	255.2352	23.64	0.41 **	2.51 **	1.91 **	2.00 **
5	Malic acid	C_4_H_6_O_5_	ESI−	133.0153	2.01	0.39 **	3.34 **	1.46	3.89 **
6	Geranylgeranylcysteine	C_23_H_37_NO_3_S	ESI−	406.2457	21.05	1.38	0.58 *	1.10	0.42 **
7	Thymine	C_5_H_6_N_2_O_2_	ESI−	125.0366	3.94	0.44 **	1.83 **	2.36 **	1.39 **
8	Tetracosahexaenoic acid	C_24_H_36_O_2_	ESI−	355.2675	22.99	0.41 **	1.63 **	1.92 **	1.33 *
9	Leucine	C_6_H_13_NO_2·_	ESI−	130.0885	2.86	0.57 **	1.59 **	1.61 *	1.39 *
10	Aspartic acid	C_4_H_7_NO_4_	ESI−	132.0315	1.35	0.42 **	2.20 **	2.61	1.56 **
11	Stearic acid	C_18_H_36_O_2_	ESI−	283.2668	25.55	0.26 **	3.93 **	5.24 **	4.88 **
12	Succinic acid	C_4_H_6_O_4_	ESI−	117.0202	2.62	0.51 **	2.08 **	2.10 **	1.71 **
13	Linoleic acid	C_18_H_32_O_2_	ESI−	279.2355	22.55	0.63 **	1.90 **	1.82 **	1.51 *
14	Uracil	C_4_H_4_N_2_O_2_	ESI−	111.0210	2.18	0.71 **	2.62 **	2.82 **	1.95 **
15	Ornithine	C_5_H_12_N_2_O_2_	ESI−	131.0837	1.27	0.69 **	1.54 **	1.68 **	1.38 **
16	Adrenic acid	C_22_H_36_O_2_	ESI−	331.2672	23.52	0.70 **	1.61 **	1.69 **	1.25 *
17	Hypoxanthine	C_5_H_4_N_4_O	ESI−	135.0322	2.24	0.75 *	1.61 *	2.35 **	1.35 *
18	Histidine	C_6_H_9_N_3_O_2_	ESI−	154.0635	1.35	0.57 **	1.57 **	2.02 **	1.64 **
19	lysine	C_6_H_14_N_2_O_2_	ESI−	145.0995	1.28	0.68 **	1.49 **	1.62 **	1.32 **
20	Phenylalanine	C_9_H_11_NO_2_	ESI−	164.0733	5.36	0.71 **	1.47 **	1.64 **	1.30 **
21	Oxidized glutathione	C_20_H_32_N_6_O_12_S_2_	ESI−	611.1507	2.75	2.29 *	1.35	2.30 *	0.45 *
22	Proline	C_5_H_9_NO_2_	ESI+	116.0709	1.51	0.67 **	1.33 **	1.72 **	1.19 **
23	Valine	C_5_H_11_NO_2_	ESI+	118.0864	2.14	0.67 **	1.52 **	1.81 **	1.28 **
24	Niacinamide	C_6_H_6_N_2_O	ESI+	123.0552	2.22	0.66 **	1.35 **	1.55 **	1.15 **
25	Pipecolic acid	C_6_H_11_NO_2_	ESI+	130.0864	1.27	0.60 **	1.08	1.39 **	1.11 *
26	Isoleucine	C_6_H_13_NO_2_	ESI+	132.1018	2.45	0.53 **	1.62 *	1.84 *	1.28
27	Urocanic acid	C_6_H_6_N_2_O_2_	ESI+	139.0499	4.14	0.49	4.33 **	6.60 **	2.03
28	Spermidine	C_7_H_19_N_3_	ESI+	146.1649	1.20	0.76 **	1.57 **	1.59 **	1.42 **
29	Glutamic acid	C_5_H_9_NO_4_	ESI+	148.0606	1.45	0.76 **	1.32 **	1.50 **	1.17
30	Methionine	C_5_H_11_NO_2_S	ESI+	150.0583	1.68	0.69 **	1.29 **	1.66 **	1.13 *
31	Phenylpyruvic acid	C_9_H_8_O_3_	ESI+	165.0543	1.87	0.57 **	1.19	1.13	1.31 **
32	Methylphenylalanine	C_10_H_13_NO_2_	ESI+	180.1015	7.19	0.75 **	1.83 **	2.62 **	1.84 **
33	Leucylproline	C_11_H_20_N_2_O_3_	ESI+	229.1545	6.48	0.47 **	1.28	1.75 **	1.82 **
34	Palmitic amide	C_16_H_33_NO	ESI+	256.2634	24.32	1.29	0.78	0.67 **	0.54 **
35	Octadecylamine	C_18_H_39_N	ESI+	270.3158	17.57	1.25	0.57 **	0.81	0.53 **
36	Linolenic acid	C_18_H_30_O_2_	ESI+	279.2321	21.35	1.19	0.65 **	0.78 *	0.60 **
37	Linoleamide	C_18_H_33_NO	ESI+	280.2640	20.87	1.69 **	0.65 **	0.66 **	0.64 **
38	Oleamide	C_18_H_35_NO	ESI+	282.2799	15.37	0.67 **	0.97	0.95	1.15 *
39	Sphingosine	C_18_H_37_NO_2_	ESI+	300.2903	21.07	1.80 **	0.44 **	0.49 **	0.53 **
40	Sphinganine	C_18_H_39_NO_2_	ESI+	302.3061	14.95	2.06 **	0.54 *	0.50 **	1.02
41	Arachidonic acid	C_20_H_32_O_2_	ESI+	305.2479	22.16	1.34	0.60	0.62 **	0.57 **
42	Linolenoyl ethanolamide	C_20_H_37_NO_2_	ESI+	324.2907	20.06	1.59 **	0.59 **	0.71 *	0.61 **
43	Oleoylethanolamide	C_20_H_39_NO_2_	ESI+	326.3062	21.62	1.70 **	0.39 **	0.44 **	0.44 **
44	Stearoylglycine	C_20_H_39_NO_3_	ESI+	342.3009	23.09	2.32 **	0.51 **	0.47 **	0.62 **
45	DGLEA	C_22_H_39_NO_2_	ESI+	350.3067	20.84	1.56 *	0.43 **	0.51 **	0.53 **
46	Taurodeoxycholic acid	C_26_H_45_NO_6_S	ESI+	500.3052	13.72	2.39 **	0.75	0.42 **	0.75
47	N-Palmitoyl phenylalanine	C_25_H_41_NO_3_	ESI+	404.3140	23.02	0.30 *	4.27 **	3.70 *	5.07 **
48	LysoPC (O-18:0/0:0)	C_26_H_56_NO_6_P	ESI+	510.3935	21.12	1.79 **	0.84	0.91	0.61*
49	LysoPC (18:3)	C_26_H_48_NO_7_P	ESI+	518.3233	16.59	0.46 **	1.89 **	1.93 **	2.19 **
50	LysoPC (18:0/0:0)	C_26_H_54_NO_7_P	ESI+	524.3726	21.10	2.43 **	0.59 **	0.52 **	0.76
51	LysoPC (20:5)	C_28_H_48_NO_7_P	ESI+	542.3234	16.94	0.34 **	2.25 **	2.19 **	3.12 **
52	LysoPC (20:4)	C_28_H_50_NO_7_P	ESI+	544.3394	18.15	0.23 **	2.61 *	2.20 **	3.60 **
53	LysoPC (20:3)	C_28_H_52_NO_7_P	ESI+	546.3572	20.01	0.29 *	5.79 **	5.44 **	4.81 **
54	LysoPC (20:0/0:0)	C_28_H_58_NO_7_P	ESI+	552.4033	20.70	2.41 **	0.40 **	0.54 **	0.44 **
55	LysoPC (22:4)	C_30_H_54_NO_7_P	ESI+	572.3703	20.44	0.51	1.64	1.74	3.70 **

RT, retention time; +, metabolites identified in positive mode; −, metabolites identified in negative mode. Metabolites identified in both positive and negative modes; * *p* < 0.05, ** *p* < 0.01. M/C: model vs. control; AEE/M: AEE vs. model; ASA/M: aspirin vs. model; Eug/M: eugenol vs. model.

## Data Availability

The data that support the findings of this study are available from the corresponding author upon reasonable request. Some data may not be made available because of privacy or ethical restrictions. The raw proteome data can be found on the iProX (https://www.iprox.cn/, accessed 4 February 2024) with the identifier (IPX0008121000).

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
