# Peer review of "Multi-Omics Approaches for Liver Reveal the Thromboprophylaxis Mechanism of Aspirin Eugenol Ester in Rat Thrombosis Model"

_ijms, 2024, doi:10.3390/ijms25042141_

Round 1

Reviewer 1 Report

Comments and Suggestions for Authors

The article describes the use of a potential natural treatment that could enhance the effects of other pharmacological treatments, with the advantage of producing fewer side effects. Moreover, it is a very comprehensive study by integrating three different omics analyses. I have some minor comments, questions and suggestions for the authors:

- How many SCX fractions did the authors collect? How were the fractions combined prior to LC-MS analysis?

- Why did the authors use a different threshold to select up and down-regualted DEPs?

- I would recommend the authors to include a supplementary table with the list of proteins quantified in the proteomics experiment and the number of peptides correspponding to each protein.

- Please improve quality of Figures 1, 5, 6, 7, 8 and Figure S3.

- Please add the name of the protein instead of the abbreviation the first time it is mentioned in the text.

- Raw data from proteomics analysis should be available in a repository (PRIDE, ProteomeExchange..).

Author Response

The article describes the use of a potential natural treatment that could enhance the effects of other pharmacological treatments, with the advantage of producing fewer side effects. Moreover, it is a very comprehensive study by integrating three different omics analyses. I have some minor comments, questions and suggestions for the authors:

  1. How many SCX fractions did the authors collect? How were the fractions combined prior to LC-MS analysis?

In this study, 15 SCX fractions were collected. Prior to MS analysis, the fractions were not combined, and respectively analyzed by Q-Exactive MS.

  1. Why did the authors use a different threshold to select up and down-regualted DEPs?

Threshold for selecting up and down-regulated DEPs was same in the manuscript. Compared with the model, the proteins with a P value less than 0.05 and fold changes > 1.2 (up-regulated proteins) or < 0.83 (down-regulated proteins) were considered as DEPs for further analysis.

  1. I would recommend the authors to include a supplementary table with the list of proteins quantified in the proteomics experiment and the number of peptides correspponding to each protein.

Following the suggestion, we provided a supplementary table related to quantitative proteins to be placed inside the supplementary material.

  1. Please improve quality of Figures 1, 5, 6, 7, 8 and Figure S3.

Following the suggestion, we improved the quality of the pictures and put them in a folder.

  1. Please add the name of the protein instead of the abbreviation the first time it is mentioned in the text.

Following the suggestion, we made changes as followed:

fibrinogen gamma chain (Fgg), fibrinogen Alpha Chain (Fga), fibrinogen beta chain (Fgb), complement C6 (C6), complement C1s (C1s), G protein subunit alpha i1 (Gnai1), G protein subunit gamma 12 (Gng12), diacylglycerol kinase theta (Dgkq), protein tyro-sine phosphatase non-receptor type 1 (Ptpn1), transferrin (Tf), lithogenic gene 4 (Lith4), syndecan 4 (Sdc4), syndecan 1 (Sdc1), EH domain containing 2 (Ehd2), orosomucoid 1 (Orm1), hemopexin (Hpx), and kininogen-2 (Kng2).

  1. Raw data from proteomics analysis should be available in a repository (PRIDE, ProteomeExchange.)

We have uploaded raw data from proteomics analysis. The raw proteome data can be found on the iProX (https://www.iprox.cn/) with the identifier (IPX0008121000).

Reviewer 2 Report

Comments and Suggestions for Authors

This study uses transcriptomics, proteomics, and metabonomics to determine the effects of aspirin eugenol ester (AEE) in a rat liver thrombosis model. The methods and results are generally well described. This manuscript can benefit from English language editing.

1. It is well established that aspirin inhibits platelet aggregation. However, the proteomic analysis reported only significant effects on “platelet activation, signaling, and aggregation” by AEE, not aspirin (as shown in Figure 3 and Table 2). Explain why this might be.

2. In the Discussion lines 438-446, it is confusing to read and keep track of when results from prior studies and when results from the current study are being considered. The should be rewritten to clarify what each study shows and whether arachidonic acid is increased or decreased in various conditions.

3. Table 3, line 41 appears to indicate that arachidonic acid levels were lower in the aspirin group compared to the model group. However, this is not indicated to be statistically significant although it appears that it should be, if the authors could please clarify.

4. Please mention limitations in the discussion.

5. As the discussion is quite long, a “Conclusions” section is needed to summarize the main findings. The last paragraph of the Discussion section provides an inadequate summary. It is overly repetitive, repeating the phrase “antithrombotic effects” several times, without mentioning specific findings to support these conclusions.

Comments on the Quality of English Language

Proofreading required.

Abbreviations such as DGE and DEP in the abstract should be explained at first use.

On lines 24-25, “tricarboxylic acid cycle e (TCA cycle)” should be rewritten as “tricarboxylic acid (TCA) cycle.”

On lines 26-27, “[…] inhibiting platelet activation, anti-inflammation, and regulating amino acid and energy metabolism” should be rewritten as “[…] inhibiting platelet activation, decreasing inflammation, and regulating amino acid and energy metabolism.” This is in order to maintain parallel structure, e.g. https://owl.purdue.edu/owl/general_writing/mechanics/parallel_structure.html

On line 40, the word “efficiently” should likely be “efficacy”.

On lines 169-170, “different abound proteins (DEPs)” should be “differentially expressed proteins (DEPs).”

On lines 305-306, “DEPs” is in a larger font than the surrounding text.

On line 432, “94” should be written as “Ninety-four” since it starts a sentence.

The abbreviation list at the end of the manuscript is incomplete.

Author Response

This study uses transcriptomics, proteomics, and metabonomics to determine the effects of aspirin eugenol ester (AEE) in a rat liver thrombosis model. The methods and results are generally well described. This manuscript can benefit from English language editing.

  1. It is well established that aspirin inhibits platelet aggregation. However, the proteomic analysis reported only significant effects on “platelet activation, signaling, and aggregation” by AEE, not aspirin (as shown in Figure 3 and Table 2). Explain why this might be.

Figure 3 showed that AEE was only strong for protein interactions in platelet activation, signaling, and aggregation. The differential protein analysis showed that ASA was also enriched for platelet aggregation, but this is not shown in the figure. We placed the protein analysis data inside the supplementary material to further support our results. Table 2 results showed that ASA was not as effective as AEE in inhibiting the expression of thrombus-related proteins. This may be related to Eug, and AEE exerted the common effect of ASA and Eug.

  1. In the Discussion lines 438-446, it is confusing to read and keep track of when results from prior studies and when results from the current study are being considered. The should be rewritten to clarify what each study shows and whether arachidonic acid is increased or decreased in various conditions.

We made changes as followed:

Arachidonic acid is associated with many physiological and pathological processes, especially inflammatory responses. The plasma metabolomics results showed that the arachidonic acid content was higher in ASA group compared to the carrageenan group [23]. In contrast, this study showed that arachidonic acid levels were higher in liver of the model group than in the control group, while ASA group could reverse this phenomenon. The above-mentioned phenomena may be due to the reduced synthesis of arachidonic acid in the liver, but it accumulates in the blood due to the inhibition of cyclooxygenase activity by ASA, which prevents the conversion of arachidonic acid to prostaglandins. Amino acid catabolism releases energy. This study showed that some amino acid levels were significantly lower in the model group than in the control group, while the ASA group could upregulate them.

[23] Ma, N., Yang, Y., Liu, X., Li, S., Qin, Z. & Li, J. (2020) Plasma metabonomics and proteomics studies on the anti-thrombosis mechanism of aspirin eugenol ester in rat tail thrombosis model, Journal of proteomics. 215, 103631.

  1. Table 3, line 41 appears to indicate that arachidonic acid levels were lower in the aspirin group compared to the model group. However, this is not indicated to be statistically significant although it appears that it should be, if the authors could please clarify.

Because of the larger within-group differences in metabolite arachidonic acid levels in the ASA group, Standard Deviation was larger. Therefore, they were not statistically significant from each other. We put the raw data inside the supplementary material to further support our results.

  1. Please mention limitations in the discussion.

This study also has some limitations, for example, the experimental results were not systematically validated; a single animal model and cross-validation of multiple animal models were not performed.

  1. As the discussion is quite long, a “Conclusions” section is needed to summarize the main findings. The last paragraph of the Discussion section provides an inadequate summary. It is overly repetitive, repeating the phrase “antithrombotic effects” several times, without mentioning specific findings to support these conclusions.

AEE could modulate the expression of multiple genes, protein abundance, and metabolite levels in rat liver in the κ-carrageenan-induced rat tail thrombosis model. A total of 132 differentially expressed genes, 159 differentially expressed proteins, and 55 differential metabolites were identified. AEE could reverse to some extent the changes in them caused by κ-carrageenan. Some DEPs associated with thrombosis were analyzed by PRM, including Fga, Fgg, Fgb, or m1, Hpx, and kng 2. Compared with the control group, the expression of all these proteins was upregulated in the model group. However, AEE reversed the up-regulation of these proteins and also regulated the expression of metabolites, bringing them back to normal levels. Therefore, AEE had good preventive effects from thrombosis. As precursor drugs of AEE, both ASA and Eug had certain prophylactic effects on thrombosis, but the prophylactic effect of AEE was superior to that of ASA and Eug. AEE could prevent thrombosis by inhibiting platelet activation, reducing inflammation, and regulating amino acid and energy metabolism. We will perform multiple thrombus models to systematically validate the efficacy of AEE in preventing thrombosis and will explore the mechanism of AEE in preventing thrombosis from the perspective of platelet activation by whole transcriptomics and proteomics techniques for platelets.

  1. Abbreviations such as DGE and DEP in the abstract should be explained at first use.

Following the suggestion, we made changes as followed:

differentially expressed genes (DGEs)

differentially expressed proteins (DEPs)

  1. On lines 24-25, “tricarboxylic acid cycle e (TCA cycle)” should be rewritten as “tricarboxylic acid (TCA) cycle.”

Following the suggestion, we made changes.

  1. On lines 26-27, “[…] inhibiting platelet activation, anti-inflammation, and regulating amino acid and energy metabolism” should be rewritten as “[…] inhibiting platelet activation, decreasing inflammation, and regulating amino acid and energy metabolism.” This is in order to maintain parallel structure, e.g. https://owl.purdue.edu/owl/general_writing/mechanics/parallel_structure.html

Following the suggestion, we made changes.

  1. On line 40, the word “efficiently” should likely be “efficacy”.

Following the suggestion, we made changes.

  1. On lines 169-170, “different abound proteins (DEPs)” should be “differentially expressed proteins (DEPs).”

Following the suggestion, we made changes.

  1. On lines 305-306, “DEPs” is in a larger font than the surrounding text.

Following the suggestion, we made changes.

  1. On line 432, “94” should be written as “Ninety-four” since it starts a sentence.

Following the suggestion, we made changes.

  1. The abbreviation list at the end of the manuscript is incomplete.

Following the suggestion, we made changes as followed:

AEE: Aspirin eugenol ester.

ASA: Acetylsalicylic acid.

EUG: Eugenol.

DGEs: Differentially expressed genes.

DEPs: Differentially expressed proteins.

TCA: Tricarboxylic acid cycle.

CMC-Na: Carboxymethylcellulose.

NASID: nonsteroidal anti-inflammatory drug.

LC-MS: Liquid chromatography-mass spectrometer.

PCA: Principal component analysis.

OPLS-DA: Orthogonal partial least squares discriminant analysis.

VIP: Variable importance for projection.

BP: Biological process.

PRM: parallel reaction monitoring.

Round 2

Reviewer 2 Report

Comments and Suggestions for Authors

All comments have been adequately addressed. No new concerns.